# Security Requirements Prioritization Techniques: A Survey and Classification Framework

**Shada Khanneh and Vaibhav Anu ***

Department of Computer Science, Montclair State University, Montclair, NJ 07043, USA
* Correspondence: anuv@montclair.edu

**Abstract:** Security requirements Engineering (SRE) is an activity conducted during the early stage of the SDLC. SRE involves eliciting, analyzing, and documenting security requirements. Thorough SRE can help software engineers incorporate countermeasures against malicious attacks into the software's source code itself. Even though all security requirements are considered relevant, implementing all security mechanisms that protect against every possible threat is not feasible. Security requirements must compete not only with time and budget, but also with the constraints they inflect on a software's availability, features, and functionalities. Thus, the process of security requirements prioritization becomes an integral task in the discipline of risk-analysis and trade-off-analysis. A sound prioritization technique provides guidance for software engineers to make educated decisions on which security requirements are of topmost importance. Even though previous research has proposed various security requirement prioritization techniques, none of the existing research efforts have provided a detailed survey and comparative analysis of existing techniques. This paper uses a literature survey approach to first define security requirements engineering. Next, we identify the state-of-the-art techniques that can be adopted to impose a well-established prioritization criterion for security requirements. Our survey identified, summarized, and compared seven (7) security requirements prioritization approaches proposed in the literature.

**Keywords:** software engineering; requirements prioritization; software security; requirements engineering





## 1. Introduction

Security, in the context of information technology services and software engineering, is becoming a topic that is garnering tremendous attention. The degree of ubiquity and availability the world is witnessing in software driven services, networking, and shared resources is recognizably higher in recent years. With that in consideration, security presents itself as an integral part in developing a successful and reliable software system that incorporates the necessary measures for protecting stakeholders' assets [1]. Requirements Engineering (RE), which is the earliest and an essential stage of software development lifecycle (SDLC), presents software developers with the opportunity to identify and document security requirements for the software-being-built.

Security is a vast and complex concept that addresses within its folds many constraints and quality aspects, including privacy, confidentiality, integrity, availability, and interoperability. Security concerns must be addressed and accounted for in the early stages of the software development lifecycle (SDLC) to avoid serious security faults in the system [2–8]. Furthermore, a clear process must be established and tailored for analyzing and addressing security requirements. The current practice in software development projects barely allocate time to address the subject of security requirements analysis in the Requirements Engineering (RE) phase of SDLC and instead push any security events analysis to the next SDLC phase (i.e., the software design phase). However, leading researchers in the software engineering literature recognized the need to address security concerns more adequately in the RE phase itself so as to prevent security risks, attacks, and financial loss [2,4,9–12]. For

example, Mead et al. [11] reflected based on NIST reports, that software that is faulty in security and reliability costs the United States economy alone around $59.5 billion annually in breakdowns and repairs. The costs of poor security requirements show that there would be a high value to even a small improvement in this area. After an application is deployed and is in its operational environment, it is difficult and expensive to significantly improve its security. Bearing in mind these elements, security requirements engineering and elicitation, was granted a specific area of knowledge in the information technology literature.

In view of security requirements, it naturally comes to mind that it is non-negotiable and must be considered valuable. Furthermore, while all security requirements are relevant and must be accounted for, the software industry is highly competitive and fast-paced. Software engineers are bound to work with the constraints of schedule and budget. When it comes to security requirements, they must compete not only with the limited resources such as time, budget, restricted human power, etc., but also, with the constraints they inflict on a software-application's availability, features, and functionalities. Therefore, security requirements prioritization is an extremely valuable activity. A well guided prioritization technique can provide substantial guidance for software engineers to make educated decisions on which security requirements are of topmost importance, which in turn guarantees that at least the most critical security protection mechanisms are addressed. Thus, it is essential to have a clear and comprehensive analysis process and selection criteria to determine which security requirements are of topmost priority in early software releases, and which can be either adopted later or not at all [3,12–19].

Security requirements engineering (SRE) is a time-intensive process that entails the need to identify, analyze, and prioritize security requirements. In addition, while one might argue that using well established requirements engineering and prioritization techniques could be applied to security requirements (SR), there is no doubt that a need still exists to alter these techniques to fit the reach and specificity of security requirements. Park et al. [20] state that prioritization of requirements is often used in software engineering processes, but the same methodologies cannot be used with efficiency when dealing with security requirements because there are additional elements that are unique to security.

The current paper presents a literature survey approach to define security requirements engineering in the context of software applications and services with a focus on the state-of-the-art techniques, currently available to impose a well-established prioritization criterion for security requirements (SR). The work presented in this paper, aims to provide a summary of such techniques to aid software engineers determine the prioritization approach that fits the specific nature of their software system's security requirements, all while balancing other constraints such as time, budget and quality.

The rest of this paper is organized as follows. Section 2 summarizes similar work. Section 3 provides a background on security requirements engineering and prioritization. A description of currently available prioritization techniques for security requirements (SR) is provided in Section 4. A discussion of the findings of this work is presented in Section 5. Conclusions and future work are described in Section 6.

## 2. Related Work

There exists some research related to security requirements engineering (SRE) that mainly addresses the process of security requirements gathering (i.e., elicitation). It is notable that there is a lack of surveys (and literature reviews) that specifically focus on security requirements prioritization. However, this section highlights some work carried out on surveying literature that focuses on establishing a unified definition of security requirements and the techniques used for its elicitation.

In the context of defining security requirements engineering and the techniques used for their elicitation, Tondel et al. [21] conducted a literature survey to identify and describe concrete techniques for eliciting security requirements. Concrete, in this context refers to techniques that can be recognized as simple, suitable, and immediate enough to be used by ordinary software development projects (where security is not the most pressing

concern). In search of these techniques, the authors derived a summary of some major approaches to security requirements engineering in relation to tasks recommended as part of the requirements phase. This led the authors to recognize that identifying threats, assets, and security objectives, are the most recommended security requirements tasks. A lightweight approach was then created and described to be convenient enough-due to its lightweight- for average software developers to adopt for identifying and eliciting the most critical set of security requirements. Some other prioritization techniques are briefly touched upon in this article in Section 3.3. It should be noted that identifying the "most critical" security requirements (using a well-defined prioritization technique and process) remains somewhat vague and subject to the software developers' skills and experience.

Previous research work also highlighted a lack of a unified and universally accepted definition of security requirements (SR), and an agreement within the software community on what is an SR [21]. This issue of multiple definitions of security requirements in literature, was recognized by Haley et al. [22] and Turpe [23] as well. In addition to the lack of consistency and the difficulty to understand satisfaction criteria, and how to derive security requirements from business goals, Haley and his colleagues [22] proposed a framework to address these needs, which provided a definition of security requirements derived from the functional requirements of a system, and the security goals in operational terms. This definition emphasizes on security requirements as constraints on the functional requirement, and not as constraints themselves. In addition, the definition requires security requirements to express the system's security goals in operational terms, precise enough to be given to a designer/architect. This notion advocates that security requirements, such as functional requirements, are prescriptive, providing a specification (behavior in terms of phenomena) to achieve the desired effect [22]. Salini et al. [6] seem to have accepted and adopted this definition for SR. The authors carry on with that definition to establish the activities that each security requirements engineering (SRE) process must account for as an essential step in the process. These are: (a) Assets, threats, and vulnerabilities identification. (b) Threat modeling (c) Risk analysis to prioritize the identified SR (d) Security requirements specification using a security requirements specification language or modeling to remove the identified errors (e) Requirements' specification review/inspection. As a result of this work the authors identified 11 SRE methods and derived a comparison based on how these methods define SR and how much of the specified activities they cover. As a conclusion the authors state that, based on their literature survey security requirements should be considered as functional requirements. Additionally, they believe that SQUARE and Security Requirements Engineering Process methods cover most of the important activities of SRE. Thus, developers can adopt these SRE methods and easily identify the security requirements for software systems.

With regards to the techniques used for SRE, researchers have conducted systematic literature reviews to summarize literature related to SRE in light of what techniques are available for the process of its elicitation. Mellado et al. [12] carried out a systematic review concerning security requirements engineering in order to summarize the evidence regarding this issue and to provide a background to appropriately position new research activities. Anwar Mohammad et al.'s [2] systematic review work had a more specific agenda of summarizing existing security requirements engineering (SRE) approaches to derive a comparison along with the best practices adopted in the field of SRE. Another systematic mapping study was established by Villamizar et al. [8] to outline the literature stand on the issue of security requirements engineering in Agile environments. The authors inferred that due to the conflict of concept between the philosophy of agile methodology and that of SRE, a challenge arises on how to incorporate SRE in an agile environment.

However, none of these systematic reviews define a question to distinguish literature and techniques that cater for the prioritization of security requirements (SR). It is also worth mentioning here that none of the above-mentioned work focuses on surveying, analyzing, and/or describing security requirements (SR) prioritization techniques. To that end, in our

work we provide a comprehensive review of the published literature that addresses the area of security requirements prioritization.

### 3. Definitions and Background

This section first defines Requirements Engineering (RE), followed by a discussion on Security Requirements Engineering (SRE) in the context of software development process. Finally, a discussion on security requirements prioritization is provided.

#### 3.1. Requirements Engineering in Software Development

The software development life cycle or SDLC (shown in Figure 1) refers to a framework where the process of software development is divided into stages as follows: requirements engineering, design, implementation, testing, maintenance and support. Requirements Engineering has been regarded as an essential step that must proceed the implementation and development of the software's architecture and code [24]. By the way of definition, Requirements Engineering (RE) refers to the process of gathering/eliciting, analyzing, documenting, and maintaining requirements in the software engineering process [25]. It has been established that most of the work related to "correct requirements definition" and "requirements fault removal" is carried out in the early stages of the SDLC. This is due to the fact that addressing requirements errors, such as ambiguous, incomplete, or omitted requirements, is more expensive to fix later in the software lifecycle. However, it is important to note that requirements usually change and evolve during development as well as after a system has been operational for some time, which is an important factor to be considered in change management activities [16,25,26]. Figure 1 demonstrates the RE process with respect to the SDLC.

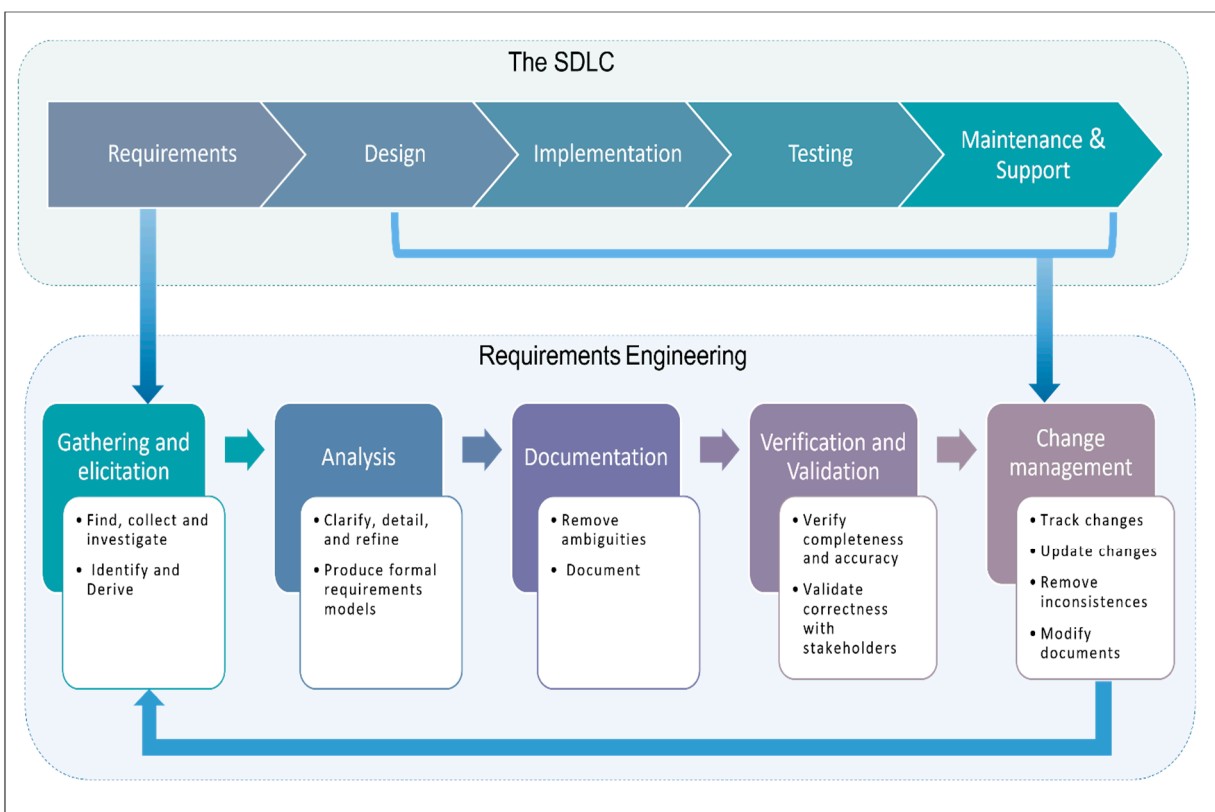

**Figure 1.** Requirements Engineering Process (as a part of the SDLC).

One of the most essential activities, carried out in the analysis phase of RE, is producing the requirements specification for the software-to-be. Traditionally, a part of this analysis is to distinguish the functional from the non-functional requirements. Functional

requirements are those that can directly relate to an action the software-to-be is expected to perform as per the end-user expectations. Nonfunctional requirements, on the other hand are more ambiguous to derive and normally refer to those requirements that express a software-to-be's constraint or quality, such as performance, reliability, efficiency, and security [26]. Security requirements, specifically, have been subject of controversy with regards to their classification as a functional or non-functional requirement (this will be further discussed in the next section). Table 1 provides examples of functional and non-functional requirements to clarify the differences between the two concepts.

**Table 1.** Examples of Functional and Nonfunctional Requirements.

| Example Software Requirement | Requirement Category |
|---|---|
| Providing users with the capability to upload files | Functional requirement: allows for file upload action |
| Ability to process large files of up to 1 Gb | Nonfunctional requirement: performance, load |
| Users should receive emails of new merchandises | Functional requirement: allows for email sending action |
| The system should issue an email within two seconds of adding new merchandise to the database | Nonfunctional requirement: performance, efficiency |
| Display warning messages and instruction messages to the user | Functional requirement: allow for messages display action |
| Messages should be brief, clear, and readable in plain English without disclosing any sensitive information. Error messages should be colored red. Instruction messages should be colored green. | Nonfunctional requirement: reliability, user experience, security. |

*3.2. Security Requirements Engineering*

Security requirements (SR) is usually categorized under the general concept of nonfunctional requirements. SRs are still recognized by leaders in the software engineering industry as a constraint of a software system. This common definition is crucial and has placed security requirements at the back of the queue when selecting requirements that must be carefully considered and implemented (especially in the software's first and early releases). The competitive nature of software-based services locks designers and business providers in a state where the delivery of functional requirements that satisfy the stakeholders needs as soon as possible is of topmost priority. For decades, the focus has been on implementing as much functionality as possible before the project deadline and patching the inevitable bugs when it is time for the next release or hot fix [2,21,27]. However, security breaches are notably increasing exponentially with the connectivity and extensive use of information systems and are more likely to compromise software-applications today [2]. This makes security one of the most predictable concerns. It also enforces a shift in how to consider and define security requirements. The literature recognized this dire need to better define security requirements in a manner that brings it to the attention of software engineers and developers.

The definition of security requirements (SR) must be adjusted to place it in its rightful place as a necessity that stakeholders must not easily compromise or disregard. On that notion, Haley et al. [22] derived a framework to define security requirements as a constraint on the functions of the system, where these constraints operationalize one or more security goals. In addition, while this definition still regards security requirements as nonfunctional requirements. It does, however, expand the reach of SRs by demonstrating how they must operationalize the security goals. Anwar Mohammad et al. [2] reflect that in the light of this definition, security requirements are the security goals that have to be satisfied in the context of the system's environment rather than implementations of security measures

and policies. They are dynamic in nature and evolve as the software is developed. The reason that most of the researchers classify them as non-functional requirements is because they do not have a clear-cut yes/no criteria for their specification and satisfiability. In this regard, security as constraints, are preventative measures that express the system's security goals in operational terms, precise enough to be given to a designer/architect. Furthermore, security requirements must be prescriptive, providing a specification (behavior in terms of phenomena) to achieve the desired effect [22]. This is very similar to how functional requirements are defined and expected to behave in software engineering. Kobilica et al. [4] add that, although security belongs to a class of non-functional requirements (NFRs) related to system dependability, many security requirements are functional in nature. Perhaps this is why Salini et al. [6] took a more biased stand and concluded that security requirements should be considered as functional requirements. Moreover, recently, researchers have argued that considering security requirements along with functional requirements tends to integrate security solutions into the system early, thereby avoiding serious flaws in the final software product [2].

Defining security requirements purely as functions introduces a different argument of misconceptions that might jeopardize the value of security requirements elicitation process. Security requirements when expressed as standalone functions, could omit significant information to the design mechanism used for implementing a specified security requirement. Firesmith [9] cautioned about confusing security requirements with the architectural security mechanisms that are used to fulfill them as it could lead to unnecessarily constraining the security team from using the most appropriate security mechanisms for meeting the true underlying security requirements. Defining requirements in terms of function distances key information such as what objects need protecting and, more importantly, why the objects need protecting [22]. Tondel et al. [21] extend on Firesmith's [9] advice on how to avoid binding security requirements to a design mechanism by focusing on defining the who, what, when, and why rather than the how. Anwar Mohammad et al. [2] came to a similar conclusion that, to develop a secure system, software engineers must first clarify and decide what to secure, against whom, why and up to what extent of security is needed. Table 2 demonstrates examples of security requirements and how they can be expressed as functional or nonfunctional requirements.

**Table 2.** Examples of Security Requirements.

| Security Requirement as Functional Requirements | Security Requirements as Nonfunctional Requirements |
| --- | --- |
| System must notify users when there is a breach. | The system must protect the integrity of personal data to prevent leakages and unauthorized access, exposure, replication, usage, or tampering |
| All passwords must be encrypted/hashed when transmitted over the network and when stored in the database | The system must be consistent where information stored matches information provided to users |
| All input fields must be sanitized and validated | System assets must be available for authenticated and authorized users only. |

Even though researchers have different opinions on whether it is more beneficial to define security requirements as functional requirements or as constraints that achieve a goal, they agree on the importance of security requirements for software project success. Overall, the literature agrees and emphasizes on the necessity of incorporating Security Requirements Engineering (SRE) principles into each phase of the software development lifecycle (SDLC). Most importantly, security requirements must be defined and analyzed in the early stages, i.e., the requirements gathering and elicitation stage [2–8]. As mentioned earlier, the ubiquity, complexity, and excessive use of software applications is also what exposes them to more security threats. Software engineers can no longer afford outdated

practices of cobbling up security features after the software is developed, where vulnerabilities are identified at a later stage and then patched after penetration testing (this is a reactive approach and not prudent). Rather, a proactive approach is needed in the current day scenario, where vulnerabilities along with the possible threats are identified, mitigated, and resolved at the early stages of the SDLC.

*3.3. Security Requirements Prioritization*

Security requirements engineering (SRE) is still a vague and cumbersome process with different considerations and approaches. As mentioned earlier, the literature has no dearth of practices that software engineers can use to adequately analyze security requirements. Tondel et al. [21] listed nine security requirements analysis approaches discussed in literature. Amongst these nine frameworks only three include categorization and prioritization as a step or a task to be carried-out in the process of eliciting security requirements. Other work presented in literature, extended some of these nine approaches or other approaches to cater for a prioritization technique. For example, Yoo et al. [19] extended the misuse-cases solution and enhanced it to incorporate a method for prioritizing security requirements, while addressing the problem of optimal risk management.

However, this absence of a prioritization step in engineering security requirements in several approaches invokes the question: is it worth the effort? Or does this absence contribute to the mistake of neglecting security requirements? Security is rarely at the forefront of stakeholder's concerns during software development, except perhaps to comply with basic standards or legal requirements [12]. Security requirements benefits are normally operating behind the scenes. Especially when compared to attractive, tangible, fast, and easy to implement features and functionalities that interact with stakeholders and end-users directly, security requirements fall victim to an emphasis on implementing functional requirements. Furthermore, security requirements must also compete with budget and time. Even though all security requirements are relevant and essential to the successful implementation of a reliable system, adopting and implementing all security requirements is onerous and almost impossible [13–15,19].

Prioritization is a common strategy adopted in software development to identify the most valuable requirements so they can be implemented in the first releases. It is also suitable to define which security requirements should be implemented first, or even which ones are mandatory or optional depending on the context. This is a pragmatic strategy to incrementally incorporate security aspects into the product and to create a financial and operational plan, in which the most important requirements are handled first, so the high-risk threats and issues are addressed as soon as possible [7,13]. Prioritizing security requirements helps in determining the topmost critical security concerns that must be addressed and implemented in the system's early releases. Moreover, when considered carefully, prioritization supports other tasks performed in SRE. For example, adopting a SRE method with prioritizing security requirements in mind or as a goal enforces the consideration of threats impact, the value of the assets being protected, the vulnerabilities under question, and implementation cost to outcome value mapping. In addition, the process of prioritizing security requirements emphasizes and sheds light on their importance, thus solidifying their chances of being adopted and selected among other competing requirements.

Ideally prioritization must be incorporated in the requirements engineering (RE) stage of the SDLC. The absence of a properly tailored requirements prioritization process could arguably lead to software development taking more time than it should (when compared to a software development process supported by adequate prioritization). Most current literature draws focus on tasks covering the definition of what a security requirement is and concepts that correlate to it such as, assets, objects, threats, and vulnerabilities. However, without proper prioritization the establishment of these definitions alone does not support time-efficient software development. The optimal purpose of security requirements is to support the objectives of a project and the overall value of a project's quality and success.

What prioritization offers in terms of risk-analysis and trade-off-analysis is valuable for stakeholders and software engineers to make educated trustworthy decisions regarding: which security requirements are more critical and implemented first when developing a software.

### 3.4. Terms and Definitions

For the reader's convenience, Table 3 provides definition of important concept that are frequently mentioned in the upcoming sections of this paper.

**Table 3.** Security Requirements Prioritization: Important Terms and Definitions.

| Term/Acronym | Definition |
| --- | --- |
| Asset | An object in the system that holds value to the stakeholders, whether it is tangible (e.g., cash, hardware, people) or intangible (e.g., information or reputation) [3,28]. |
| CRAMM | CRAMM stands for Central Computing and Telecommunications Agency (UK) Risk Analysis and Management Method. A method used to calculate the measure of risk for each threat to an asset and vulnerability [3,29]. |
| DREAD | An algorithm used to compute a risk value, as an average of five categories [3]:DREAD = (Damage + Reproducibility + Exploitability + Affected Users + Discoverability)/5. |
| Risk | A future condition or circumstance that exists outside the control of the project team that will have an adverse impact on the project if it occurs. While an issue is a current problem that must be dealt with, a risk is a potential future problem that has not yet occurred [30]. |
| Security Bugs | A vulnerability in the software that might lead to security attacks (e.g., Denial of service attacks, remote code execution, unauthorized access). |
| Threat | A hypothetical event that has the potential to cause some performing damage to an organization's business and other processes. Threats mostly do not cause any damage unless they are being actualized and exploited by malicious actors. |
| UML Modeling | The Unified Modeling Language is an industry-standard graphical language for specifying, visualizing, constructing, and documenting the artifacts of software systems (e.g., Use case diagram, class diagram, sequence diagram, misuse cases diagram, threat/attack trees) [24,29]. |
| Vulnerability | A weakness in the system that an attack exploits. Vulnerabilities enable attackers to actualize and exploit a threat. In other words, a feature or property of a system that helps an attack to succeed. The more vulnerable a system is, the higher is the expected success rate of attempted attacks and the more choices an attacker has. Examples of software vulnerabilities include but are not limited to poor programming practices (e.g., unsanitized user input, failure to check array bounds), poor authentication and authorization process, uncontrolled network access, human and environmental factors [23]. |
| STRIDE | An acronym for Spoofing identity, Tampering with data, Repudiation, Information disclosure, Denial of service, and Elevation of privilege. STRIDE is a popular approach used for threat modeling and classification [21,31]. |

### 3.5. Research Methodology

The rest of this paper (mainly Sections 4 and 5) provide the results of a comprehensive literature survey that we performed. This sub-section provides an overview of the research methodology used including paper selection The main goal of our survey was to identify, summarize, and classify the security requirement prioritization techniques proposed in the literature. The research methodology is described as follow:

- *Search Query*: We used the search query "security requirements prioritization" for our search process.
- *Research Databases Searched*: We ran our search query on ACM Digital Library, IEEExplore Digital Library, and Google Scholar databases.
- *Inclusion-Exclusion Criteria:* We only selected those papers that focused on proposing and empirically evaluating a security requirements prioritization technique. We excluded papers where the main language was not English. We also excluded papers where full text not was not available.

## 4. Security Requirements Prioritization Techniques: Developing a Classification Framework

Security requirements engineering, elicitation, and prioritization is a multidimensional and complex task. Achieving it efficiently while maintaining a balance between constraints, represents a challenge for software engineers. Leaders and keen researchers in the discipline of software engineering have recognized this difficulty. To alleviate this challenge, many techniques were synthesized to guide the process of security requirements engineering (SRE) and prioritization. The focus of this section is to introduce some of these techniques and assess their validity and capability as presented and evaluated in literature.

In this work we have identified seven (7) techniques in published literature where a prioritization technique for security requirements was described, implemented, and evaluated on a real-world software example. Table 4 provides a summary and examination of these techniques. Please note that in Section 4.1 through Section 4.7 we provide a more detailed insight into each of these techniques. Additionally, based on our observations we provide a classification scheme (illustrated in Figure 2) to help the readers better understand each technique and where it is more suitable and applicable based on the software system size and nature. Each security requirements (SR) prioritization technique was classified under one of the following software project scales:

- *Small Scale*: projects that have no, or few identified risks, are closed in less than a year, have a low relative budget, easy to carry out and complete (e.g., proven, and reliable technology, existing site, off the shelf software).
- *Medium Scale*: projects with some identified risks, take at least a year to close, have a considerable work effort and budget, have identified changes and difficulties (e.g., Involves newer technologies, integrated to other operations in the organization).
- *Large-to-Mega-Scale*: projects that have multiple identified risks and challenges, take at least two-years to meet the definition of carried out, have relatively high budget, require expertise and training to carry out (First of kind technology, different organizational approach or new direction, change in vision or industry).

**Table 4.** Summary of Security Requirements Prioritization Techniques.

| Technique | Prioritization Deterministic Factors | Key Advantages | Key Limitations |
|---|---|---|---|
| Technique 1: Park et al. [20] Treat Modeling and Valuation Graph | <ul><li>The value of assets, threats, and countermeasures.</li><li>The impact: the level of harm an attack inflicts on an asset. In terms of that impact's, damage, recoverability, and likelihood.</li><li>TCO (total cost of ownership) for each countermeasure.</li><li>TI, total impact of all threats a countermeasure mitigates</li><li>The gain: TCO − TI</li><li>SR priority = Gain + asset value</li><li>Asset value, and the countermeasures TCO are used to determine the order for SR of the same priority.</li></ul> | <ul><li>Accounts for business users' feedback on defining assets.</li><li>Accounts for developing, training, and maintaining countermeasure cost (TCO).</li><li>Provides clear guidelines to establish a reasonable prioritization technique. While leaving room for flexibility.</li><li>Suggests a solution for SR with similar priorities.</li><li>Provides graphical representation</li></ul> | <ul><li>Does not provide a definitive clarification of vulnerabilities and does not account for their valuation.</li><li>Does not define the business goal a SR is trying to achieve.</li><li>Was evaluated on a small example. The graph might be hard to track for large project representation.</li></ul> |

**Table 4.** *Cont.*

| Technique | Prioritization Deterministic Factors | Key Advantages | Key Limitations |
|---|---|---|---|
| Technique 2: Gulati et al. [3] Framework | • Risk value for threats, vulnerabilities, and assets<br>• Risk estimate = using CRAMM based on threats, vulnerabilities, and assets values.<br>• SR priority = sum of all prioritized threats based on their risk estimate. | • Defines systems vulnerabilities<br>• Accounts for variabilities valuation.<br>• Characterizes threats according to STRIDE, which helps non-technical better stakeholders relate these threats to their needs. | • Tight dependency on external methods (CRAMM, DREAD, STRIDE) which could compromise this approach's ability to be generic and easily adaptable.<br>• Multi-layer of factors mapping could make this approach hard to follow and understand.<br>• Some ambiguities in derivation and valuation of vulnerabilities, impact of assets, and threat prioritization.<br>• Does not suggest a solution for SRs that have the same priority.<br>• Does not account for SR implementation cost.<br>• Does not define the business goal a SR is trying to achieve. |
| Technique 3: Yoo et al. [19] Enhanced Misuse-Case | • Risk of Misuse-case (harmful actions) based on CVSS method.<br>• Total risk of a misuse-case: the risk of a use case multiplied by the number of use-case (functional assets) it harms.<br>• SR priority = sum of the total risks for each misuse-case this SR mitigates.<br>• The number of misuse-cases a SR mitigates determines the order of similar priority SR. | • Incorporates why an attacker wants to harm a system by defining goals, giving more realistic meanings to security threats.<br>• Extends on a popular system modeling method (the use-case UML diagram), which makes it simpler to adopt.<br>• Directly correlates the SR to functional requirements.<br>• Fairly easy calculation process.<br>• Offers a solution for SR with similar priority.<br>• Provides graphical representation | • Does not define and value, assets, vulnerabilities, SR implementation cost<br>• Does not define the business goal a SR is trying to achieve.<br>• Was evaluated on a small-scale example.<br>• Tightly coupled to the use-case diagram and functional requirements; functional requirements are always changing thus any change or adding of functional requirements requires the whole method to be reconstructed and the risk to be reevaluated. Will perform poorly in agile environments. |
| Technique 4: Sharma and Ajit Framework [7] | • Assets, vulnerabilities, and SR implementation cost.<br>• The cost of the damage a vulnerability inflicts on the system's assets.<br>• The priority of each vulnerability in terms of the difference between the total possible damage cost to each asset and the implementation cost to remove errors which cause a vulnerability occurrence.<br>• SR Priority = sum of the priority values of each vulnerability mitigated by this SR. | • Addresses the issue of identifying information related directly to the organization and the environment the system will operate within.<br>• Accounts for implementation cost.<br>• Could be used as a standalone prioritization scheme for SR that are already identified and established.<br>• Could be used as a generic guideline. | • There is not a clear distinction between vulnerabilities and threat, the two concepts seem to be addressed as one in this approach.<br>• Variabilities are expressed as threats not as the underlying weaknesses in the system.<br>• Was evaluated on a small-scale example. |

**Table 4.** *Cont.*

| Technique | Prioritization Deterministic Factors | Key Advantages | Key Limitations |
|---|---|---|---|
| Technique 5: SQUARE Process [11] | • Artifacts, business goals, risk assessment of impact and likelihood of threats affecting organization's risk tolerance. | • Accounts for business goals<br>• Provides a generic guideline and process to follow.<br>• Provides a checklist for software engineers to ensure incorporating important aspects. | • The success of this method relies on the skills and expertise of the team using it.<br>• The process is fixed where each step is required for the next one. Might be hard to follow for agile fast based environments.<br>• The evaluation was established on fielded systems. The performance of this process engineering a project in its infancy is still vague and undetermined. |
| Technique 6: AHP [32] | • Pairwise relative value of the SR<br>• Pairwise Relative cost for implementing each requirement<br>• Cost to value mapping = SR priority | • Accounts for the security requirement relative value in comparison to other SR.<br>• Accounts for implementation cost.<br>• Includes visual representation.<br>• Can be easily used as a stand-alone prioritization technique after establishing the SRs.<br>• Useful for determining the priority of security requirements that have the same value. | • Low flexibility: changing one SR requires the revaluation and calculation of all security pairs.<br>• Might be strenuous for large-scale projects to compare and calculate all pairs of requirements.<br>• Missing many considerations in prioritizing SR; the underlying mitigated threats, vulnerabilities, and business goals.<br>• Not many examples and case studies are available in literature where AHP was used to specifically prioritize security requirements. |
| Technique 7: Carvalho et al. [13] Risk assessment, the AHP method and Generic scenarios approach | • Severity of threats.<br>• Severity of security issues in terms of regulations and standards.<br>• AHP result for SR with same value.<br>• SR priority = threat severity + security issue severity. | • Accounts for the security requirements importance regarding the standards and regulation it addresses.<br>• Provides a solution for SR with similar priority values.<br>• Can be appended as a standalone technique for prioritizing established SR.<br>• Incorporating the bug-bar for severity calculations and the AHP method adds robustness to the prioritization result. | • Does not account for implementation cost, business goals, and stakeholders needs.<br>• Inherits some difficulties and rigidness from calculating the risk values based on the bug-bar and the AHP pairwise method. |

### 4.1. Prioritization Technique 1: Threat Modeling and Valuation Graph

Proposed by Park et al. [20] this approach facilitates the threat modeling model to create a process that allows for the prioritization of security requirements via the valuation of assets, threats, and countermeasures. Modeled in a tree-like structured graph referred to as a "valuation graph".

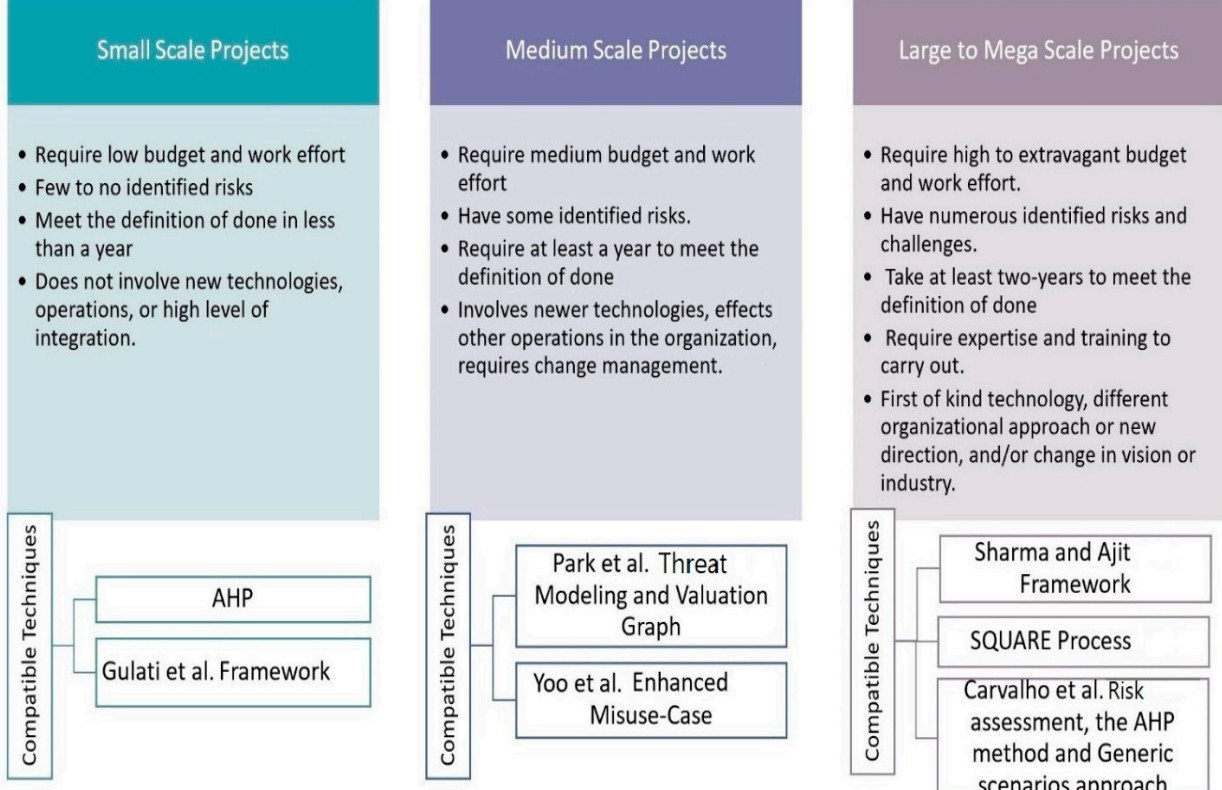

**Figure 2.** Security Requirements Prioritization Techniques: A Classification Framework with Respect to Project Size.

The valuation graph requires a total of eight steps: six steps that must proceed the prioritization scheme in order to achieve the prioritization, which is manifested in seventh and eighth steps. These steps are as follows:

1. *Identify assets*: in this step, a list of all critical assets of the system is created. In this approach the authors stress the importance of deriving this list not only depending on management decisions but must also account for business users' needs.

2. *Draw and value assets*: once the list of assets requiring protection is recognized, the initial part of the valuation graph can be created. A tree where the system is the parent node, and the assets are the children. Additionally, this step includes an essential output that must be generated to perform the prioritization step more adequately. Which is the value of each asset. The authors suggest using any valuation method then expressing this value as a number between 1 (low importance) and 5 (high importance).

3. *Identify threats per asset*: in this step, the most important task is to adopt the mindset of an attacker, identify possible points of attack, and analyze the vulnerabilities of exposed assets. The authors suggest using approaches such as vulnerabilities lists or STRIDE to derive these threats.

4. *Draw threats and calculate their impacts*: once threats have been identified, they are drawn in the valuation graph using one node per threat and joining them to the assets via edges. As for calculating the impact, it is expressed as the level of harm that an attack can cause to an asset when the threat is exploited (Impact = (Damage + Recoverability + Likelihood)/3).

5. *Identify countermeasures*: once the possible threats have been recognized, it is necessary to identify countermeasures. In this step the authors suggest approaches such as misuse cases and attack trees to diagram and analyze the relation between threats, vulnerabilities, and countermeasures.

6. ***Draw and value countermeasures***: once the countermeasures are identified, they are drawn in the valuation graph via new nodes between assets and threats, joining the assets that are protected and the threats that are mitigated. When a threat is related to various countermeasures, it is necessary to clarify whether one of the countermeasures is enough to mitigate the threat, or whether all or parts of the countermeasures are necessary to mitigate the threat. Additionally, each countermeasure value is calculated to determine their cost. The authors highlight that when this value is measured, it is not enough to calculate the development cost of the countermeasure. However, it is also necessary to calculate the user training cost and maintenance cost. This implies a calculation of the total cost of ownership (TCO). Finally, once each countermeasure's TCO is calculated, this value is transformed into a number between one and five, where five represents the highest cost of ownership. After this transformation, the values are entered into the valuation graph.

7. ***Calculate priority of countermeasures***: security requirements become the countermeasures that were synthesized in previous steps. In addition, using the output of the six previous steps the security requirements (SR) can be prioritized as follows:

   (i)    Calculate the total impact (TI) of threats that a countermeasure mitigates. TI is the summation of the impacts of each threat related to the countermeasure. If a threat is related to more than one countermeasure, its impact is divided by the number of edges if the relation is an AND; the impact is summed if the relation is an OR.

   (ii)   Calculate the gain (G) of each countermeasure by subtracting the countermeasure TCO from its TI.

   (iii)  Calculate the priority (P) of the countermeasure by summing the (G) of the countermeasure and the value of assets that the countermeasure protects.

8. ***Sort countermeasures***: Once the priorities of each countermeasure have been calculated, they are sorted according to priority. If two or more countermeasures have the same priority, the asset value, and the countermeasures TCO are used to determine the order.

As for evaluation, the authors [20] applied the proposed approach steps to an example of an e-commerce web application that controls the bill-payment data of customers and the catalog of products sold on this site. For this example, three business logic were used as the assets. Six threats to these assets were synthesized. Five countermeasures were identified as the elicited security requirements (SR) and then prioritized according to steps seven and eight of the proposed approach. Examining the steps required to accomplish this prioritization scheme. It is clear that the authors have set clear guidelines for software engineers to follow. Many considerations regarding best practices and important aspects of security requirements were brought to light. For example, assigning a value to the assets to derive more meaning for the SR, calculating the cost for implementing the SR, while also accounting for training cost and maintenance cost, deriving the impact as a result of damage, recovery, and likelihood instead of just the damage, and using AND/OR relationships to clarify whether one of the SR is enough to mitigate the threat, or whether all or parts of the SR are necessary. However, as of all techniques it is hard to cater for all aspects of security. With that, the most obvious limitation of this approach is the absence of a definitive expression of vulnerabilities and their valuation. This is critical for deciding which variabilities are causing the most harm to a system. Which could affect the value of the importance of a countermeasure, and the underlying protection mechanism. For example, knowing how many threats are imposed due to a said vulnerability in a system. Could affect the priority of the security requirements that address the threats imposed by this particular vulnerability. Additionally, the approach was evaluated on a small-scale example, making it hard to assess its performance for real world applications. Where expectedly, the graph will be complex to plot and derive meaningful observations from it.

*4.2. Prioritization Technique 2: Threat Analysis and Risk Measurement*

Gulati et al. [3] presented a technique for prioritizing security requirements (SR) based on the threat analysis risk measurement technique. The authors suggest eight steps to adequately achieve this prioritization technique.

1. ***Assemble threats***: using common criteria-based approach list all the possible threats to develop a storage of deposits of all these threats.
2. ***Characterize all the known threats using STRIDE***: to classify the schemes for characterizing the discovered or known threats according to the kinds of exploitation that are used.
3. ***Rate the assembled threats using DREAD methodology***: this algorithm is used to compute the risk value of a threat, as an average of five categories (DREAD = (Damage + Reproducibility + Exploitability + Affected User + Discoverability)/5).
4. ***Assigning the values to vulnerabilities***: the authors suggest the CRAMM method to accomplish this step. The CRAMM calculates the measure of risk for each threat to an asset and vulnerability.
5. ***Define and value assets***: define the project's specific assets. Then assign a value to each by weighing the impact of it when a threat will occur.
6. ***Calculate risk level***: risk is defined as the probability that a threat agent will exploit system vulnerability and thereby create an effect detrimental to the system.
7. Risk = Value based on Measure of Threat, Vulnerability and Asset.
8. ***Find the security requirement to lessen the threats***: define security requirements as the countermeasures that lessens a threat's risk.
9. ***Backtrack the security requirements prioritizations***: backtrack all the gathered values of different threats which are discovered earlier and assign them a final value which will prioritize all the security requirements.

To evaluate this proposed framework the authors [3] conducted a case study on an example of an air reservation system. Seven threats were assembled. Five assets were identified for this example. The example does not illustrate what vulnerabilities were defined and the authors only state that the values for the vulnerabilities were defined by the CRAMM. This approach introduces important measures to be accounted for and incorporates various methods. Which could be beneficial for software engineers who are familiar with and using them in their SRE process. In such a case, it could be simpler to extend such a process to offer prioritization by appending this approach. On the other hand, this tight dependency on these methods (CRAMM, DREAD, STRIDE) compromise this approach's ability to be a generic and smoothly adaptable approach that can be tweaked for an existing process. Additionally, this multi-layer of mapping made the approach hard to follow and understand. Finally, many concepts were introduced with ambiguity and without specific guidelines on how to establish their values. For example, vulnerability is defined as the weakness in the system that makes an attack more likely to succeed. However, the authors did not give any guideline on how to express these vulnerabilities before assigning a value to them. In addition, no example of a possible system vulnerability was given in the case study. Another example is, when describing, and demonstrating step five "define the project's specific assets. Then assign a value to each by weighing the impact of it when a threat will occur", it is not clear how to define and weigh this impact.

*4.3. Prioritization Technique 3: Enhanced Misuse-Case*

The enhanced misuse-case suggested by Yoo et al. [19] extends upon the well-established use-case diagram. Which is perhaps what is most appealing about this approach. The use-case diagram is a popular UML model widely used among software engineers. In a survey asking 374 software professionals regarding security requirements engineering common practices conducted by Elahi et al. the majority of respondents (73%) expressed that they use standard or tailored modeling notations in RE practices. Where, 57% use some UML models to express the requirements, and 68% use non-UML models. In essence the use-case model

illustrates the actions and behaviors a system must allow authorized users to perform. However, and by definition, it is only concerned with the functional requirements of a system, making it non-usable for eliciting and analyzing security requirements. Software engineers recognized this limitation and extended the use-case model to describe the actions a system must not allow using what is known as the "misuse-case" model [19].

Yoo et al. discuss the misuse-case model as a solution for eliciting security requirements and describe the traditional approach to be limited for deriving security requirements that are meaningful to stakeholders. The authors conducted an enhanced misuse-case model to address some of the traditional approach's limitations and extended it to cater for a prioritization criterion and the issue of risk management.

The authors addressed the following limitations in the traditional misuse-case method are: (a) Does not provide meaningful security requirements (SR) in terms of the attacker goal, (b) Does not account for that misuse severity and impact and does not showcase any information in regard to that on the diagram, (c) Does not cater for a prioritization technique for the elicited SR.

The proposed solution suggests the following steps to address these limitations:

1. Derive the misuse case as a result of a mis-actor goal. The misuse-case is then linked to each use-case (functional requirement) it targets with a "Threatens" relationship.
2. Calculate the risks for each misuse case using The Common Vulnerability Scoring System (CVSS). The weight of that risk for each misuse-case is then adjusted according to the number of use-case it threatens.
3. Synthesize a SR for each misuse-case and link to it with a "mitigates" relationship.
4. Link each SR to each use-case it's protecting with an "includes" relationship.
5. Showcase the calculated risks and mitigations numeric values on the diagram for stakeholders to observe.
6. Prioritize each SR based on the number of misuse-cases it addresses and the amount of risk it mitigates in accordance with that calculated risk for each misuse-case.

The authors [19] advocate that with this approach a misuse-case is easier to derive and has more meaning to stakeholders. Since in this approach, one needs to think about what mis-actors want to abuse and gain, accordingly the misuse-case will be a question of how the mis-actors will achieve these goals. Additionally, calculating the risk of each misuse-case in terms of its severity and the effect it inflicts on a said system functional requirement, integrates more value to that misuse-case in the context of the environment and emphasizes it to stakeholders. Finally, as for the prioritization technique the same concepts apply. Each SR will be prioritized based on its ability to lessen the risk of each threat as well as the number of threats it is addressing.

As for evaluation, Yoo et al. [19] applied the steps they proposed to elicit and prioritize security requirements (SR) for a simple example of an e-commerce web application system. The example contained five use-case (functional requirements). The end result yielded a total of three SR that were prioritized and showcased on a diagram. With that example, it is unclear how this proposed system will scale for large-scale real-world applications where many and changing functional requirements exist. Specially for agile environments and businesses that adopt the agile methodology where functional requirements are changing even more frequently. Additionally, many considerations regarding SR are not accounted for nor addressed. Calculating the priority on severity and probability alone eliminates other aspects that matter for software engineers and stakeholders. For example, cost and time to implement overall outcome value and cost reduction mapping, its importance to stakeholders, and will it constrain and hinder the availability and performance of said functional requirements, if yes to what extent.

### 4.4. Prioritization Technique 4: Hybrid Security Requirement Prioritization Framework

Sharma and Ajit [7] recognized security requirements prioritization as an integral process that must be included in the process of security requirements engineering (SRE). The authors explain that, despite that, many approaches have been introduced for SRE. These

techniques have got certain pitfalls imbibed in them such as inefficient and inappropriate requirement gathering prioritization and hike in the specified project budget that leads to degradation in the software quality and security. To address this lacking, the authors proposed a novel framework for security requirement prioritization. The proposed framework is described by the author as a "Hybrid Security Requirement Prioritization framework". The aim of this solution is to develop a well-defined process for SR prioritization that improves the security in software applications of the business environment by gathering the properly processed requirements, identifying the vulnerabilities and their corresponding threats. Which in turn, reduced the estimated budget of the software application along with the security implication.

The proposed approach promotes that security requirements should be in sync with functional requirements and hence are required to be captured along with. Security requirements should be accurate, adequate, absolute, and non- conflicting with other requirements. Once they have been explicitly specified, they can then be implemented and maintained. Furthermore, the authors [7] state that SR should be associated with assets that must be protected and managed.

The proposed framework incorporates the following steps as part of the security requirements (SR) prioritization process:

1. Use the workshop-based approach to gather information regarding the environment or the organization the system will be operating on; (a) Important assets and their relative values. (b) Perceived threats to the assets. (c) Security requirements. (d) Organizational vulnerabilities.
2. Establish the relationship between vulnerability and error; gather and list all the vulnerabilities a system might have and the respective error that might cause a system to have these vulnerabilities.
3. Identify the relationship between vulnerability and Security Requirement. This is so that after establishing a ranking scheme for these vulnerabilities, they can be removed from the SR accordingly.
4. Define the relationship between assets and vulnerability. In terms of (a) The impact of the vulnerability over the assets. (b) The potential damage cost to each asset caused by vulnerability occurrence. (c) Define the implementation cost.
5. Calculate the total damage cost inflected on assets by a vulnerability occurrence.
6. Prioritize the vulnerabilities based on the difference between the total possible damage cost to each asset and the implementation cost to remove errors which cause a vulnerability occurrence.
7. Prioritize each security requirement based on the sum of the priority values of all vulnerabilities, corresponding to that security requirement.

The proposed framework was implemented on a case-study of an online banking system simple example. For the system six assets were identified, four vulnerabilities and five security requirements (SR). According to the priority calculation for each vulnerability and the number of vulnerabilities each SR addresses, these SR were prioritized. Basically, this approach derives the SR's priority by first prioritizing all vulnerabilities addressed by this SR. Second, the sum of the priority value for all of these vulnerabilities will be used to prioritize the SR. Although, the proof of concept was applied on a small example, making it hard to assess its validity for practical applications. The approach incorporates solid concepts to prioritize SR. Starting the process with a solid understanding and investigating the organization and the environment the system will operate within. Insurance a more comprehensive coverage of what a SR should account for. This could also reduce unnecessary efforts to address irrelevant SR to the operational environment and the organization's needs. Another beneficial aspect with this process is defining the vulnerabilities of the system and establishing a relationship between these vulnerabilities and the SR. Which could be very useful for designing the appropriate mechanisms to address such variabilities. Finally, this approach accounts for important values specially for business analysts, such as the cost of the overall damage to the organization's assets, the damage to specific assets,

and the implementation cost of the SR. However, and what is worth mentioning here is the proposed approach focuses on deriving the prioritization process for SR that are already defined and listed. The elicitation of the SR itself was not well addressed or showcased by the authors of this framework. Another consideration while examining the given case study. There was not a clear distinction between vulnerabilities and threats, the two concepts seem to be addressed as one in this approach. In addition, the given examples of variabilities seem to be addressing threats not the underlying weaknesses in the system.

### 4.5. Prioritization Technique 5: SQUARE Process

The SQUARE (Secure Quality Requirements Engineering) process seems to be highly regarded among researchers. Many praised this method to cover most of the important tasks for eliciting security requirements (SR) [2,6,21,28]. For example, Salini et al. [6] in their work of surveying SRE techniques, advocated that SQUARE and Security Requirements Engineering Process methods cover most of the important activities of SRE. In addition, developers can adopt these SRE methods and easily identify the security requirements for software systems. In a systematic review to identify and compare SRE methods. Anwar Mohammad et al. [2] recognized SQUARE to be one of the most popular amongst researchers.

SQUARE is a model developed at Carnegie Mellon by Nancy Mead as part of a research project with Donald Firesmith, and Carol Woody of the Software Engineering Institute. The focus of this methodology seeks to build security concepts into the early stages of the development lifecycle [11].

SQUARE method advocates nine steps to be used alongside existing lifecycle models to ensure adequate support for SRE:

1. Agree on definitions.
2. Identify security goals
3. Develop supporting artifacts
4. Perform risk assessment.
5. Select elicitation techniques.
6. Elicit security requirements.
7. Categorize requirements.
8. Prioritize requirements.
9. Inspect requirements.

The square method addresses the prioritization of security requirements (step-eight) in terms of many aspects provided by previous steps. These aspects are considered inputs to this step. The SQUARE framework is similar to a waterfall approach where each step is essential input to determine the shape of the next step's outcome. Basically, however, the factors that must be established in order to determine the priority of a said system SR, are the list of artifacts that needs protecting, categorized to meet the organization's established business goals. In addition, the risk assessment of how the combination of impact and likelihood of various threats affect the organization's risk tolerance with regard to each categorized requirement.

To evaluate this framework the initial SQUARE model was tested by graduate students at Carnegie Mellon University in 2004 in two consecutive case studies. Carnegie Mellon students, under the mentorship of Nancy Mead, partnered with an IT firm, Acme Corporation, to apply the model to one of the firm's fielded systems [11]. As a result of these studies the initial SQUARE model was refined to what is described as in this section. What is worth mentioning here is that the concept of SQUARE is to provide a guideline of what must be present and accounted for. While allowing to add activities to the framework as per the project needs and the stakeholders' considerations. Some limitations in regard to this methodology might stem from its similarity to the waterfall approach, and thus by default inheriting its rigid nature. Tondel et al. [21] Criticized this method in terms of its ambiguity. That stating what Square includes and does not include is not straightforward, because developers can choose several techniques for the different steps. Thus, it relies heavily on expert knowledge and the requirements team ability to facilitate the process.

Another observation regarding this approach was made by Mellado et al. [29] that the square method lacks the compliance with any Information Security Management System standard, such as ISO/IEC 17799 or ISO/IEC 27001, as well as the steps of SQUARE do not deal with the security requirements reuse.

### 4.6. Prioritization Technique 6: Analytic Hierarchy Process (AHP)

Analytic Hierarchy Process (AHP) was developed by Prof. Thomas L. Saaty and applied to software engineering by Joachim Karlsson and Kevin Ryan in 1997 [32]. AHP is widely known in software prioritization as a structured technique for organizing and analyzing requirements to be prioritized. Prioritization decisions are made based on mathematics and psychology. AHP is helpful for decision making in situations where multiple objectives are present. This method compares all pairs of requirements in order to calculate a relative value for each security requirement.

Karlsson et al. [32] describe the AHP to consist of five steps as follows:

1. Review candidate requirements carefully to test for completeness and to ensure that they are stated in an unambiguous way.
2. Apply AHP's pairwise comparison method to assess the relative value of the candidate requirements.
3. Estimate the relative cost of implementing each candidate requirement.
4. Calculate each candidate requirement's relative value and implementation cost, then plot these on a cost–value diagram.
5. Map, analyze, and discuss the candidate requirements. Based on discussion, prioritize the requirements, and decide which will be implemented.

Although, AHP was originally synthesized for prioritizing said system requirements without specifically addressing the security requirements. It is widely considered by researchers to be applicable and useful for prioritizing SR [13,18]. The biggest argument against AHP in terms of SR is that it does not provide any input on how to elicit and manufacture SR. On the other hand, it provides a solid prioritization technique once the SR are established. In addition, could be particularly useful as an add-on step to any other SRE process that does not address prioritization. For example, Sadiq et al. [33] demonstrated how AHP can be applied to prioritize SR, using a detailed example scenario. The authors also add that by using AHP, requirements engineers can confirm the consistency of the result and avoid subjective judgment errors and increase the likelihood that the results are reliable. Carvalho et al.'s Approach (discussed in next section) to identify and prioritize SR shows how AHP can be incorporated in the prioritization process to derive better results [13]. In terms of limitations that exist within the process itself, Herrmann et al. [34] observe about AHP that it does not scale well because the number of comparisons grows exponentially with the number of requirements. The authors add and further explain that requirements can be estimated in cardinal values (absolute values) or ordinal values (relative values/ratio scale). AHP belongs to the latter. When risks and priorities are quantified on a cardinal scale an existing list of prioritized requirements is more easily scalable and extensible than when ordinal values are used. New requirements can easily be inserted in the list, without the need to compare each one to the whole list of the other requirements [28].

### 4.7. Risk Assessment, the AHP Method and Generic Scenarios Approach

The work presented by Carvalho et al. [13] is mainly concerned with security issues and incidents associated with smart toys that use sensors and cloud-based services to collect data. Next, it is used to personalize the user's experience, or to perform operations such as navigation, for example. To adequately address the issues the authors proposed an approach where they used the Microsoft SDL method to identify a comprehensive list of security issues based on specific regulations, threats based on surface attack analysis, and security requirements that address security issues and threats. Finally, the authors

presented a method to prioritize security requirements based on risk assessment, AHP, and generic scenarios.

The proposed approach to prioritize security requirements after they are identified and elicited, is as follows:

1.  Identify the severity of all threats addressed by the security requirements using the security bug bar. The security bug bar is an objective bug classification system where the severity of a threat is expressed as Low, Moderate, Important or Critical.
2.  Define the severity of all security issues addressed by the security requirements. The authors identified security issues as the standards and regulations that must be addressed in the context of a system.
3.  Calculate the risk of each security requirement based on the severity of the threats and security issues addressed by it. Use this value to prioritize the SR.
4.  Find the best implementation order for security requirements with the same risk using AHP.
5.  Synthesize the prioritized list of security requirements.
6.  Check consistency of the prioritized list of security requirements.

The above-mentioned prioritization scheme was conducted as part of eliciting and synthesizing SR for smart toys. To further evaluate its performance, the authors compared the outcome of this method to that of NAT (The Numerical Assignment Technique (NAT). NAT is a straightforward way to prioritize requirements based on the customer's point of view) [13]. As a conclusion of this comparison, the authors reflect that using the bug bar and AHP technique adds advantage and robustness. In general, this proposed approach is flexible enough to be appended to other systems. However, this approach might not scale well in accordance with cost and time. Reconstructing this method each time a security issue or threat severity changes, requires recalculating the risk, and the AHP value. Additionally, other considerations regarding SR were not addressed using this approach (such as the cost to implement and the business goals).

## 5. Discussion

This survey was initiated with three main goals:

*   *Goal 1*: The first goal was to understand the stance and coverage of current literature on the concept of the Security Requirements Engineering (SRE) and elicitation process.
*   *Goal 2*: The second goal was to identify techniques that address the process of security requirements (SR) prioritization.
*   *Goal 3*: The third goal was to identify a comparison scheme to better understand the capability of each prioritization technique, and which technique would be more suitable to implement for specific software projects.

*Goal 1 observations*: We provided the literature survey on SRE in Section 3.2. It is notable that much research has been carried out on the topic of SRE. Many researchers advocate the importance of adopting the discipline of SRE as part of the early stages of the software development lifecycle (SDLC). They emphasize that in today's world, software engineers cannot afford to consider security as an afterthought. Despite this focused effort, SRE is still exhibiting symptoms of a novice concept. There are various reasons for this including: (a) The literature does not seem to agree on a unified definition of what a security requirement is. (b) There is not a clear standard of what the process of eliciting SR must include, (c) There is not a unified acceptance of how to measure the correctness of a SR once established, (d) many proposed techniques suggested to address SRE do not consider a prioritization scheme.

*Goal 2 observations*: We provided a brief review of literature on security requirements prioritization in Section 3.3. The literature provides many techniques and frameworks to guide the process of adequately eliciting security requirements. Additionally, many survey efforts were conducted to summarize, list, and compare these techniques. Researchers recognized and emphasized the importance of prioritizing SR, to help software engineers

address and derive educated decisions in the matter of risk-analysis and trade-of-analysis of competing objectives. However, many of the proposed techniques for SRE do not address prioritizing security requirements. We also noticed that there does not exist a comprehensive survey that summarizes and discusses approaches that cater for prioritizing security requirements.

With that motivation, the third part of this work aimed to cover this gap and summarize some of these SR prioritization techniques (please see Table 4). Our survey resulted in identification and summarizing of seven techniques and frameworks that are described in current literature to guide software engineers when they are trying to prioritize security requirements.

*Key observations regarding security requirements prioritization techniques*: The literature recognized the gap in addressing prioritization as an integral step during the security requirements engineering activity. This is evident by the different approaches and proposed frameworks summarized in this paper. What is interesting in regard to these techniques is the different factors and considerations each of them requires as essential to the process. For example, the enhanced misuse-case approach expresses security requirements (SR) as the countermeasures that lessens the misuse risk. This approach prioritizes each SR according to how many of the system's functional requirements it is protecting and how much risk is it reducing. While, Carvalho et al. [13] prioritized each SR based on its risk value that is calculated in terms of the severity of the threats and security issues (the standards and regulations that must be addressed in the context of a system) addressed by this SR. The common denominator in all these techniques is that one must account for risk impact when prioritizing security requirements. The issue becomes how to express this impact and on what terms. Hansch et al. [26] describe a model-based impact analysis method as a proven one to understand the security needs of complex IT systems. Using such an analysis, requirements for protective measures can be derived and prioritized.

Another observation is that many of these listed techniques require a set of essential steps that must proceed the SR prioritization scheme to establish adequate results. These preceding steps force software engineers to address and think about many important aspects of security requirements, such as the vulnerabilities, threats, cost, business goals, regulations and standards, relative value, risk impact and its probability, etc. Hence providing a more comprehensive overall SRE elicitation process and affirms the validity of these security requirements. This confirms an assumption previously made by this paper, regarding how the absence of a prioritization step could jeopardize the value of the elicitation process. In addition, deriving the elicitation process with prioritization in mind allows for many considerations to be sufficiently addressed, to establish more relevant and tangible security requirements.

*Goal 3 observations*: It is notable from the analysis carried out in our survey that each of the discussed SR prioritization techniques differ in terms of complexity, required level of experience, flexibility and resilience to requirements change, and the aspects of software security it reflects (e.g., organization's goals, assets, cost, vulnerabilities, operational environment). This variation presents somewhat of a challenge when deriving a classification schema. However, to best incorporate each variation factor, we established the classification based on the software project size and nature, where a project can either be a small-scale, medium-scale, or large-to-mega-scale project. With this project categorization we can establish the characteristics of each category based on, budget, time, and effort valuation, number of identified risks and their severity level, complexity in terms of change degree and frequency, and the maturity level of the organization. With that, each SR prioritization technique can be adequately classified under each category, such that the selected prioritization technique incorporates and accounts for the said project's needs, size, and nature.

*Application of Identified Prioritization Techniques on Modern Technologies:* In addition to the above-mentioned discussion, we also wanted to understand the application of security prioritization techniques on modern technologies such as IoT, blockchain etc. One such technology is the use of Digital Twins for securing cyber-physical systems as described

by Suhail et al. [35]. Digital Twins are virtual replicas of cyber-physical systems, which can be monitored to understand/predicting the behavior of the actual system without performing any analysis/inspection on the actual system. We propose that while developing Digital Twins, security prioritization techniques such as misuse case [19] to ensure Digital Twins consider all possible attack vectors. Another application that we envision is in the area of blockchain. Application of blockchain-based technology and relevant security considerations have been proposed and described by authors in areas such healthcare [36] and manufacturing industry [37]. Specifically, Pal [37] propose in their paper that when integrating IoT and blockchain has many data privacy challenges. Many such data privacy challenges can be revealed by techniques such as SQUARE [11].

## 6. Conclusions

This paper presented a literature survey approach to assess the current state-of-the-art on the concept of security requirements engineering and prioritization process.

The major contributions of this study include:

- A major goal of this study was to identify, gather, and present the available security requirements prioritization techniques in a single resource, so that future researchers can find most of the valuable information in a single place. To that end, this work identified, summarized, and compared seven (7) techniques depicted in literature. These techniques can help guide software engineers in better eliciting and prioritizing security requirements.

- Another major contribution of this work is that we classified the 7 security requirements techniques based on the size of the software development project (please see Figure 2 in Section 4). Based on our classification scheme shown in Figure 2, software development teams and software project managers can select the security requirement prioritization technique/framework that is most suitable for their project.

- Another major contribution of this work is to provide a comparative analysis of existing security requirements techniques so that industry software developers can choose the technique that is suitable for their specific project. Table 4 provides this comparative analysis. To our knowledge, such a comparative analysis of security requirements techniques has not been performed so far in the literature.

As a follow up to this survey, we are currently working on performing a case study on real world software requirements artifacts (i.e., software requirements specifications or SRS documents). Our plan is to compare the coverage and ease-of-implementation of each of the seven security requirements prioritization techniques on a real SRS document.

Another future work is to extend our literature survey to understand what kind of security requirements prioritization have been incorporated into some prominent and well-known requirements engineering methodologies. The examples of such methodologies include the STORE methodology proposed by Ansari et al. [38] and TOPSIS methodology [39]. Specifically, in the STORE [39] methodology, prioritization is carried out at multiple levels (stakeholder, risk, etc.) to ensure that a more sound and complete security requirements specification document is created.

Since security of cyber-physical systems as well as IoT devices has gained tremendous attention recently, in our future work, we want to analyze the kind of security requirements prioritization techniques that are relevant to risk assessment of cyber-physical systems and IoT systems. We intend to include work by Northern et al. [40] and Suhail et al. [35] in a future study on cyber-physical system security. Additionally, security requirements prioritization technique such as misuse cases and SQUARE can be incorporated in some known IoT security approaches such as the approach proposed by Andreas et al. [41] which focuses on secure healthcare data exchange.

Another future work is to interview software development teams from organization spanning two countries (USA and India). In our interview, we will collect and analyze data about the security requirements elicitation and prioritization techniques used in

real world software development projects and also train software developers on existing prioritization techniques.

**Author Contributions:** Conceptualization, S.K.; Supervision, V.A.; Writing—original draft, S.K.; Writing—review & editing, V.A. All authors have read and agreed to the published version of the manuscript.

**Funding:** This research received no external funding.

**Institutional Review Board Statement:** Not Applicable.

**Informed Consent Statement:** Not applicable.

**Data Availability Statement:** Not applicable.

**Conflicts of Interest:** The authors declare no conflict of interest.

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
