# Peer review of "Security Requirements Prioritization Techniques: A Survey and Classification Framework"

_2674-113X, doi:10.3390/software1040019_

Round 1
Reviewer 1 Report
The authors raised the crucial topic of security requirements. The article is structured correctly and the content is presented in a logically consistent order. The authors managed to propose a very interesting article.
However, in the "Discussion" section the authors may also elaborate on applying the surveyed techniques to up-to-date technologies. The authors mention Digital Twin [15], but the paper lack discussion on blockchain and IoT. Besides, the authors may comment on various industry applications. Please do consider the following papers to use in the "Discussion" section:
- "Towards situational aware cyber-physical systems: A security-enhancing use case of blockchain-based digital twins" (https://doi.org/10.1016/j.compind.2022.103699),
- "Reconfigurable Smart Contracts for Renewable Energy Exchange with Re-Use of Verification Rules" (https://doi.org/10.3390/app12115339),
- "Privacy, Security and Policies: A Review of Problems and Solutions with Blockchain-Based Internet of Things Applications in Manufacturing Industry" (https://doi.org/10.1016/j.procs.2021.07.022),
- "A Review of Blockchain-Based Secure Sharing of Healthcare Data" (https://doi.org/10.3390/app12157912).
Minor stylistic or punctuation errors should be corrected but they do not diminish the value of the manuscript.
In the "Conclusion" section, the authors should focus on the conclusions of the review. Your own input should be underlined at the end of the "Introduction" section. Please refine the conclusions.
In my opinion, it is worth publishing the proposed article in the Software journal.
Author Response
Dear Reviewer,
Thank you for your detailed review and comments on the manuscript. We really appreciate your valuable input. A major comment was to add more about up-to-date technologies in our "Discussion" section. We have added a new paragraph at the end of our discussion section (Section 5) to address this comment.
Another major comment was to refine the conclusion section. In order to address this comment, we have made major changes to the conclusion section in the updated manuscript (Section 6). We have now included a bulleted list of our major contributions.
We appreciate your time.
Sincerely,
Authors
Reviewer 2 Report
Thank you to the authors who invested their time and efforts to bring forth this paper. The study used a literature survey approach to define security requirements engineering. They also identified the state-of-the-art techniques that can be adopted to impose a well-established prioritization criterion for security requirements. Below are the comments and suggestions which, when appropriately addressed by the authors, may enhance the quality of the paper:
1. What makes this study different or unique from the rest of the previous studies? Specify the unique contributions of this paper.
2. Improve the research methodology. The type of this paper is not clear.
3. Most of the references are old. Cite some recent references.
Ansari, M. T. J., Al-Zahrani, F. A., Pandey, D., & Agrawal, A. (2020). A fuzzy TOPSIS based analysis toward selection of effective security requirements engineering approach for trustworthy healthcare software development. BMC Medical Informatics and Decision Making, 20(1), 1-13.
Fujs, D., & Bernik, I. Characterization of Selected Security-related Standards in the Field of Security Requirements Engineering.
Ansari, M. T. J., Pandey, D., & Alenezi, M. (2018). STORE: Security threat oriented requirements engineering methodology. Journal of King Saud University-Computer and Information Sciences.
Author Response
Dear Reviewer,
We are really grateful for your detailed review and constructive comments on our manuscript.
One major comment was: "What makes this study different or unique from the rest of the previous studies? Specify the unique contributions of this paper." In order to address this comment, we have expanded our conclusion section (Section 6). We have provided a bulleted list of contributions of this survey study that we consider interested researchers will find valuable.
Another major comment was: "Most of the references are old. Cite some recent references." This valuable comment has given us a new idea about how to extend our survey further for a research study. We have addressed this comment by adding a new paragraph in conclusion section (the new paragraph is the second last paragraph in conclusion section).
Another comment was about mentioning the study type. We have now mentioned in the paper that this paper follows as literature survey approach. We have also added a completely new sub-section on research methodology (sub-section 3.5) in the updated manuscript.
Sincerely,
Authors
Reviewer 3 Report
The survey is missing the methodology used for the gathering of the reference.
What was you inclusion criteria?
How you gathered the papers in the area?
What was you exclusion criteria?
Following work in the area like the following needs to be included and considered in the writing of the paper e.g. 10.1007/s11042-021-10827-x and 10.3390/info12100408.
Author Response
Dear Reviewer,
We really appreciate your thoughtful review and comments on our manuscript.
One major comment was: "The survey is missing the methodology used for the gathering of the reference.". We have addressed this comment by adding a new sub-section called Research Methodology (sub-section 3.5) to our updated manuscript. This sub-section describes our paper selection method, inclusion-exclusion criteria.
Another major comment was to include some relevant work done by Northern et al. and Andreas et al. We have incorporated these works and more in our updated manuscript.
Thank you again for your time.
Sincerely,
Authors
Round 2
Reviewer 2 Report
The authors made the requested corrections, only a review in English for small corrections is lacking. Authors must review the English, and make the necessary corrections.